# The Effect of the Use of a Settling Chamber in the Cultivation of *Penaeus vannamei* and *Salicornia neei* in Aquaponics with Bioflocs

**DOI:** 10.3390/ani15091294

**Published:** 2025-04-30

**Authors:** Isabela Pinheiro, Flávia Banderó Höffling, Felipe Boéchat Vieira, Walter Quadros Seiffert

**Affiliations:** 1Centro de Ciências Agrárias, Laboratório de Camarões Marinhos, Departamento de Aquicultura, Universidade Federal de Santa Catarina, Rua dos Coroas 503, Florianópolis 88061-600, SC, Brazil; flaviabandh@gmail.com (F.B.H.); walter.seiffert@ufsc.br (W.Q.S.); 2Alfred Wegener Institute, Helmholtz Centre for Polar and Marine Research, Am Handelshafen 12, 27570 Bremerhaven, Germany

**Keywords:** suspended solids, BFT, Pacific white shrimp, halophyte, IMTA, nutrient recycling, marine aquaponics

## Abstract

This study evaluated the impact of continuous solids removal using a settling chamber on the cultivation of *Penaeus vannamei* and *Salicornia neei* in a biofloc-based aquaponic system. Two treatments were compared: one with a settling chamber and one without. In the settling treatment, water was first directed through the chamber before reaching the hydroponic bench, while in the treatment without settling, water was pumped directly to the plants. Over a 54-day trial, the settling treatment resulted in reduced suspended solids but increased TAN and NO_2_ levels, while NO_3_ concentrations remained stable. Despite differences in water quality parameters, no significant variations were observed in shrimp or plant production between treatments. These findings indicate that the continuous use of a settling chamber is not essential for system performance throughout the 54 days, suggesting a simplified approach for biofloc-based aquaponic designs.

## 1. Introduction

The intensification of aquaculture depends on the development and application of technologies that maximize production in cultivated areas and take advantage of the reuse of water and nutrients [1]. An aquaponic system combines the components of aquaculture and hydroponics to promote the growth of aquatic organisms and plants in an integrated manner [2]. Among the advantages of the aquaponic system are the absorption of nutrients by plants, reduced environmental impacts from effluent discharge, economic diversification through the cultivation of high-value products, and increased profitability per unit of production [3,4,5].

Although freshwater aquaponics is the most practiced and studied, the limited use of water for food production and soil salinization has increased the search for alternatives, such as integrating the cultivation with halophyte plants (salt-tolerant) into brackish or marine water systems [6,7]. Halophyte species such as *Salicornia neei*—commonly known as salicornia, sea asparagus, or samphire—are of great interest in the global market for gourmet cuisine due to their mildly salty flavor and high nutritional and nutraceutical value [8,9,10].

Aquaponic systems typically consist of a tank for the rearing of aquatic organisms, a settling chamber for the removal of suspended solids, a biofilter for nitrification, and a hydroponic bed for vegetable production [11]. However, to reduce installation and operational costs and simplify system design, combining the components of solids removal and nitrification in a single unit would be advantageous [2]. One way to achieve this goal is the integration of biofloc technology (BFT) into aquaponic systems [7].

Biofloc technology (BFT) is a biosafety aquaculture system for the farming of different species and is characterized by little or no water exchange, as well as the presence of a rich microbial community responsible for controlling nitrogen compounds in water [12]. Traditional aquaponics relies on separate nitrification units, typically involving biofilters where nitrifying bacteria convert ammonia to nitrate. In a biofloc system, these nitrifying bacteria are part of the biofloc microbial community. They efficiently perform nitrification within the same system, thus integrating the process and reducing the need for additional nitrification units [13]. In whiteleg shrimp (*Penaeus vannamei*) production, BFT allows for increased stocking densities while also providing an abundant source of microbial aggregates, serving as a complementary and continuously available food source for the reared organisms, contributing to higher productivity [14,15].

Thus, the strategic implementation of integrated farming practices that include the incorporation of crops through aquaponics with bioflocs can also benefit plant growth by providing additional nutrients and improving water quality [7,13,16]. Recent studies have demonstrated that integrating shrimp and *S. neei* cultivation in bioflocs can enhance productivity and promote sustainability by enabling nutrient recycling through plant uptake [17,18,19,20,21]. The first study on biofloc-based aquaponics integrating marine shrimp and Salicornia demonstrated successful plant growth (8 kg·m^−2^) and a production ratio of 2 kg of plant biomass per 1 kg of shrimp [21]. However, if not properly managed, the high concentration of suspended solids in the water can affect the roots and impair nutrient uptake and plant respiration, leading to lower plant productivity [7].

Total suspended solids (TSS) in BFT systems arise from uneaten feed, fecal matter, debris, molted exoskeletons, and microorganisms within the culture tanks. The removal of solids with settling chambers is a common practice in *P. vannamei* cultivation in BFT [22,23]. Sedimentation occurs due to the density difference between water and bioflocs, allowing organic matter to settle at the bottom of the settling chamber through gravity [24]. However, the constant use of the settling chamber can break the solids into smaller non-settleable particles, reduce the microbial biodiversity, and remove the nitrifying bacteria biomass from the culture tanks, resulting in the instability of water quality parameters [25,26,27,28]. In aquaponic systems with bioflocs, the use of settling chambers can become practically unnecessary. Due to the high microbial activity in the mineralization of organic matter, the solids retained in the hydroponic structure are degraded, and this process releases nutrients that are absorbed by the plants, simulating natural growing conditions in the soil [2,3,16,21].

The design of aquaponic systems often varies based on production objectives and the species being cultivated [13]. To optimize performance, it is essential to understand the dynamics of solids production and manage nutrient-rich wastewater effectively, ensuring its efficient reuse for plant growth [29]. Given the importance of managing residual nutrients and protecting plant roots from possible negative effects caused by suspended solids, this study aimed to evaluate the use of a settling chamber in a biofloc-based aquaponic system for the production of marine shrimp *Penaeus vannamei* and the halophyte *Salicornia neei*, as well as to assess its effects on sludge production and water use.

## 2. Materials and Methods

The experiment was conducted over a 54-day period, from October to December 2018, at the Marine Shrimp Laboratory (LCM) of the Federal University of Santa Catarina, southern Brazil.

### 2.1. Biological Material

Juvenile *Penaeus vannamei* shrimp (Aquatec Ltda., Canguaretama, RN, Brazil) were initially reared in an intensive biofloc system (stocking density of 1000 shrimp m^−3^) with an average salinity of 33 psu.

*Salicornia neei* seedlings were propagated vegetatively through cuttings. Adult plants from the LCM plant bed were cut into 10 cm sections of the main stem, with no branches. These cuttings were planted in polystyrene trays (128 cells) using a substrate mixture of fertile soil, sand, and perlite. The trays were placed in a dark room and irrigated with tap water every two days. After 25 days, the seedlings were exposed to morning sunlight and irrigated with seawater diluted to 50% concentration every two days for another 25 days. At the end of this period, 168 seedlings were individually weighed (mean weight of 2.1 ± 0.1 g) and transferred to the aquaponic system.

### 2.2. Experimental Design and System Management

To evaluate the effects of using a settling chamber in the aquaponic system, two treatments were tested: one with settling and one without settling. Each treatment had three replicates, totaling six experimental units randomly distributed in a 243 m^2^ greenhouse.

Each aquaponic experimental unit comprised a circular polyethylene tank (800 L useful volume) for shrimp cultivation and a hydroponic bench for plant growth (Figure 1). The shrimp tanks were equipped with an 800 W titanium heater, a micro-perforated aeration hose, and four vertical artificial substrates. Each tank was covered with 50% shade netting. In the treatment with settling, a cylindrical fiberglass settling chamber with a conical bottom (90 L capacity) was attached.

The aquaponic structure was constructed based on the design by [21], with modifications. The hydroponic bench consisted of four PVC pipes (75 mm diameter, 1.10 m length) arranged side by side on wooden supports with a 4% incline. Each bench had 0.33 m^2^ of planting area, with 28 seedlings of *S. neei* per experimental unit, corresponding to a planting density of 84 plants m^−2^. In the treatment without settling, water was continuously pumped from the shrimp tank to the hydroponic bench at a flow rate of 3.0 L min^−1^ using a submerged pump (model SB650, Sarlo Better, São Caetano do Sul, SP, Brazil) and returned to the tank through gravity.

In the treatment with settling, water was pumped continuously to the center of the settling chamber, where a 100 mm PVC pipe was placed to reduce water turbulence, allowing solids to settle along the chamber’s conical bottom [2]. The settling chamber design followed the model proposed by [27]. A PVC baffle was used at the overflow outlet to prevent possible floating solids from being carried onto the hydroponic bench. Overflow water was then distributed into each irrigation channel and returned to the shrimp tank via gravity. To maintain the proper concentration of suspended solids [26], every thirty minute, the solids accumulated in the settling chamber were pumped back to the tank for 40 s using an electric pump (model EBE 01, Emicol, Itu, SP, Brazil) connected to the bottom outlet of the settling chamber, at a flow rate of 15 L min^−1^.

One day before the beginning of the experiment, the tanks and settling chambers were filled with biofloc water from the nursery tank. Initial water parameters were as follows: 356.0 mg L^−1^ of total suspended solids (TSS), 0.3 mg L^−1^ of total ammonia nitrogen (TAN), 0.1 mg L^−1^ of nitrite, 6.3 mg L^−1^ of nitrate, 1.7 mg L^−1^ of orthophosphate, a pH of 8.11, an alkalinity of 140 mg CaCO_3_ L^−1^, and a salinity of 32 psu. Each tank was stocked with 300 shrimp (average weight of 1.20 ± 0.08 g), representing a stocking density of 375 shrimp m^−3^.

The shrimp were fed four times daily with a commercial feed containing 38% of crude protein (Poti Mirim QS 1.6 mm, Guabi, Campinas, SP, Brazil), following a feeding table [30]. Calcium hydroxide was added when alkalinity was below 120 mg CaCO_3_ L^−1^, at a ratio of 20% of the daily feed intake. Throughout the experiment, organic carbon was not added, as ammonia concentrations remained below 1 mg L^−1^, indicating a predominantly chemoautotrophic system [31]. There was no water exchange, and only the volume lost through evaporation was replaced to correct the salinity. Light intensity (PPFD) was measured daily at noon above the plants (mean of 968.64 ± 388.45 μmol m^−2^ s^−1^), with a natural photoperiod of 13 h of light.

### 2.3. Production Indexes of Shrimp and Plants

After the experimental period, shrimp production indices were calculated: mean final weight (g), survival (%), weekly weight gain (g week^−1^), final biomass (g tank^−1^), productivity (kg m^−3^), and feed conversion ratio.

Plant productivity was measured by weighing the aerial portion of each plant, as root weights were not measurable because of entanglement with perlite and support screens. The mean final weight (g), the final biomass (g tank^−1^), productivity (kg m^−2^), and survival (%) were then calculated.

### 2.4. Water Quality Variables

During the experiment, dissolved oxygen and temperature were monitored twice daily (8 am and 5 pm) using an oximeter (Pro20, YSI Inc., Yellow Springs, OH, USA). Salinity (EC300A, YSI), pH (TEC-11, Tecnal, Piracicaba, SP, Brazil), alkalinity, TAN, and nitrite [32] were checked twice a week. Nitrate (method NitraVer^®^ 5, Hach, Loveland, CO, USA) and orthophosphate [32] were measured weekly.

### 2.5. Sludge Quantification

Total suspended solids (TSS), volatile suspended solids (VSS), and fixed suspended solids (FSS), and settleable solids were quantified twice weekly [33]. To maintain TSS concentrations between 400 and 600 mg L^−1^ for optimal shrimp culture [26], solids were removed as needed, and the amount of sludge was quantified [20].

Total sludge produced in each experimental unit was calculated using the following formula:(1)$$sludge\;produced\;\left(kg\;{tank}^{-1}\right)=\frac{\left(final\;TSS\times v\right)-(initial\;TSS\times v)}{1000000}+\sum SR$$
where *final TSS* and *initial TSS* represent the final and initial concentrations of the TSS (mg L^−1^), *v* is the volume in liters of each experimental unit, and *∑SR* (kg) is the total weight of sludge removed. Sludge removed from settling chambers was measured using a graduated bucket, and TSS concentrations in the removed sludge were converted to kg [34]. Sludge accumulated in the hydroponic channels was washed back into the shrimp tanks during plant harvest, and the amount is inserted in the final TSS concentration of the tank (*final TSS*).

### 2.6. Statistical Analysis

Data were tested for the normality (Shapiro–Wilk test) and homogeneity of variance (Levene’s test). Zootechnical and phytotechnical indices were compared between treatments using Student’s *t*-test. The data of sludge production were also analyzed using Student’s *t*-test. Water quality variables were analyzed using repeated measures ANOVA to assess treatment effects over the days of cultivation, followed by Bonferroni post hoc tests to correct for multiple comparisons. All tests were conducted at a 5% significance level using the software jamovi version 2.6 (The jamovi project, Sidney, Australia).

## 3. Results

### 3.1. Production of Shrimp and Plants

There was no significant difference between treatments for any variable when evaluating the productive performance of *P. vannamei* and *S. neei* (Table 1).

### 3.2. Water Quality

No significant differences were observed for salinity, temperature, and dissolved oxygen between treatments throughout the experiment. Alkalinity and pH were higher in the treatment with settling (*p* < 0.05) (Table 2). The TAN concentration was significantly higher in the treatment with settling (Figure 2a). The nitrite concentration remained constant in the treatment without settling during the experimental period and was more unstable with settling (*p* < 0.05) (Figure 2b). There was an accumulation of nitrate in the treatment without settling (*p* < 0.05) (Figure 2c). The concentration of orthophosphate increased over the weeks of cultivation in both treatments but was significantly higher in the treatment without settling (Figure 3).

### 3.3. Solids Concentration and Sludge Production

The total sludge production and the volume of settleable solids at the end of the experiment were significantly higher in the treatment without settling (Table 3). There was an accumulation of total suspended solids in the treatment without settling throughout the cultivation. In the seventh week of the experiment, the TSS concentration in this treatment was higher than 600 mg L^−1^, and a settling chamber had to be used to remove excess solids (Figure 4). Both TSS and VSS were lower in the treatment with settling (*p* < 0.05) at the end of the experiment.

## 4. Discussion

Temperature, dissolved oxygen, and salinity remained within the appropriate limits for the cultivation of *P. vannamei* and *S. neei* in aquaponics with bioflocs [18,20,21].

There was an accumulation of TSS during the experiment in the water of the treatment without settling. As it was a predominantly chemoautotrophic system, the use of the settling chamber in this treatment was only necessary almost at the end of the trial to maintain the TSS concentration within the appropriate level for the cultivation of *P. vannamei* [26], and excess solids were removed only once. In BFT systems, settling chambers are used to reduce the concentration of total suspended solids and improve the water quality and the performance of shrimp [22,26]. However, their continuous use can lead to a reduction in biofloc concentration, decrease the abundance of beneficial microorganisms, and break the microbial aggregates into small non-settleable particles, which are more difficult to control [24,25,35]. Throughout the experimental period, we observed the retention of solids in the channels of the hydroponic benches in both treatments. However, in the treatment with settling, there was also a thick layer of floating solids on the surface of the settling chambers. This was a consequence of the accumulation of fine solids, which were probably broken down during water pumping and sludge return to the tank. Thus, the reduction in the concentration of total suspended solids and settleable solids may also be responsible for decreasing the biomass of nitrifying bacteria in this treatment [1,21].

In biofloc systems, the suspended solids are formed by inorganic (FSS) and organic (VSS) components, the latter increasing due to microbiological growth [5,24]. The higher percentage of VSS in the water in tanks without settling indicates that nitrifying bacteria were more abundant in this treatment. This is also demonstrated by the greater sludge production without the use of the settling chamber, as the proper management of the solids concentration in the culture generates a more stable bacterial community [27]. In addition, in the treatment with settling, the volume of settleable solids remained close to zero throughout the entire experiment, indicating that most of the suspended solids were composed of fine non-settleable particles. In shrimp cultivation with BFT, the volume of settleable solids varies from 2 to 40 mL L^−1^ and should be maintained at around 20 mL L^−1^ [12], as observed in the treatment without settling.

The nitrification process can be influenced by a variety of parameters, such as pH, temperature, alkalinity, salinity, and the availability of oxygen and suspended solids that will serve as a substrate for nitrifying bacteria [12,31]. In the treatment with settling, the reduction in the TSS concentration led to the gradual increase in TAN and NO_2_ in the water, since lower solids levels may disrupt the nitrification process, leading to an accumulation of these compounds [28]. Furthermore, in the absence of a stable and active microbial culture, the oxidation of nitrite to nitrate occurs slowly, resulting in the accumulation of NO_2_ and the maintenance of low concentrations of NO_3_ in the crop [35,36,37]. On the other hand, the behavior of nitrite in the water of tanks without settling demonstrates that, in this treatment, the nitrification process was more stable, and NO_2_ was immediately converted to NO_3_ [38]. Consequently, there was an accumulation of nitrate, as the NO_3_ generation rate exceeded the absorption rate by the plants, which stop using this nutrient when they reach their requirement [1,38]. *Salicornia* species have been shown to grow well when supplied with either nitrate or a combination of nitrate and ammonium [39,40]. However, a balanced ammonium–nitrate ratio is recommended to maximize growth and nutrient removal efficiency [40,41], which is difficult to achieve in coupled aquaponic systems like the one used in this study. Nevertheless, the concentrations of nitrogen compounds remained below the limit tolerated by *P. vannamei* throughout the experimental period [42,43,44].

Alkalinity and pH are generally stable in tanks with bioflocs, as the nitrification process is responsible for the consumption of alkalinity and lowering the pH [12,31]. In the present study, both pH and alkalinity were higher in the treatment with settling. This may have been caused by the occurrence of denitrification in the settling chamber [22]. The accumulation of organic matter in an aerobic environment can lead to anoxic conditions in the sludge layer and promote the growth and activity of denitrifying bacteria, increasing the pH and generating alkalinity [22,45]. Moreover, these bacteria are capable of assimilating PO_4_^−3^, which would justify the lower concentration of this nutrient in the treatment with settling [27,34]. Moreover, suspended solids are the primary sources of orthophosphate accumulation, which is mainly caused by the uneaten feed. Thus, the removal of TSS in the treatment with settling may have been responsible for the lower concentration of PO_4_^−3^ throughout the experiment [46].

The zootechnical indexes were similar to those observed in integrated cultures of *P. vannamei* and *S. neei* in BFT systems [18,20,21]. Survival, productivity, and the feed conversion ratio are close to those reported by [21] and [20]. Compared to other production methods of shrimp and salicornia in aquaponics, the low mean final weight of the shrimp was probably due to the high stocking density and the scarce availability of space for the animals in the tanks. In the same production system, it was possible to harvest shrimp weighing approximately 12 g but grown at a density of 250 shrimp m^−3^ [21]. Contrary to what was observed by [22], the use of the settling chamber did not affect the productivity of *P. vannamei*, which was high in both treatments. However, further research is needed to evaluate long-term system performance as biomass and solids concentrations increase, particularly at shrimp harvest sizes of 25–30 g.

One of the concerns of growing plants in aquaponics with BFT is the management of the solids concentration in the water, as these can severely affect the roots and impact the absorption of nutrients and the availability of oxygen [7]. Though, in the present study, the continuous use of the settling chamber for the removal of TSS did not improve the production indexes of *S. neei*, which were similar in both treatments. In an experiment evaluating the integrated cultivation of *S. neei*, *P. vannamei*, and tilapia (*Oreochromis niloticus*), it was possible to produce 2.3 kg m^−2^ of plants irrigated with water with an approximate TSS concentration of 430 mg L^−1^ [20].

In an aquaponic system using settling chambers, the authors produced 1.1 kg m^−2^ of *S. neei* irrigated with marine water from the cultivation of *P. vannamei* with approximately 330 mg L^−1^ of TSS [47]. These results are close to those obtained in this experiment in the treatment without settling, which showed a higher concentration of TSS in the water throughout the cultivation. Thus, even if there is an accumulation of solids in the hydroponic channels, this can be beneficial, as the high microbial activity in the process of mineralization of organic matter releases nutrients that are absorbed by the vegetables [2,16,21]. It is worth mentioning that, although the mean final weight of the plants was lower than that obtained by other authors in similar cultivation systems [20,21], *S. neei* reached the commercial size of the *Salicornia* and *Sarcocornia* genera for the gastronomic market [6,19] before the accumulation of solids could impair the productivity of plants and shrimp.

## 5. Conclusions

This study demonstrated, under the experimental conditions presented, the viability of aquaponic cultivation of *P. vannamei* and *S. neei* in biofloc systems without the need for a settling chamber to reduce solids before the water passes through the hydroponic bench, thereby simplifying the system. Moreover, the absence of a settling chamber before the hydroponic bed resulted in more stable water quality parameters. Although the system without the settling chamber exhibited higher concentrations of suspended solids, this did not hinder the growth of either the plants or the shrimp. Nevertheless, the eventual use of a settling chamber remains necessary to maintain suspended solid levels within a range suitable for marine shrimp cultivation, particularly in aquaponic systems with longer production cycles.

## Figures and Tables

**Figure 1 animals-15-01294-f001:**
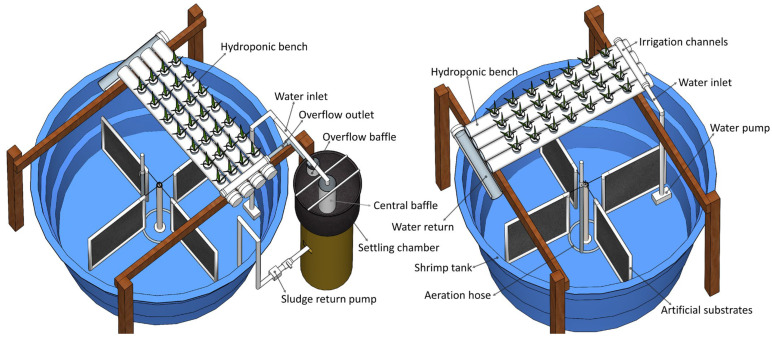
Aquaponic experimental unit used in the treatment with settling (**left**) and without settling (**right**). Adapted from [21].

**Figure 2 animals-15-01294-f002:**
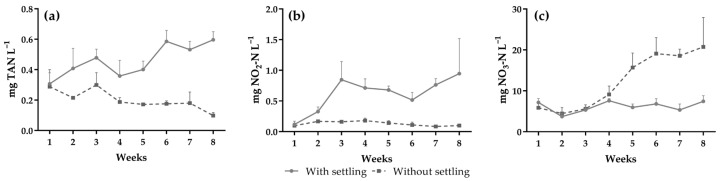
(**a**) Total ammonia nitrogen, (**b**) nitrite, and (**c**) nitrate in the tanks of *Penaeus vannamei* grown in an aquaponic system with and without the use of the settling chamber.

**Figure 3 animals-15-01294-f003:**
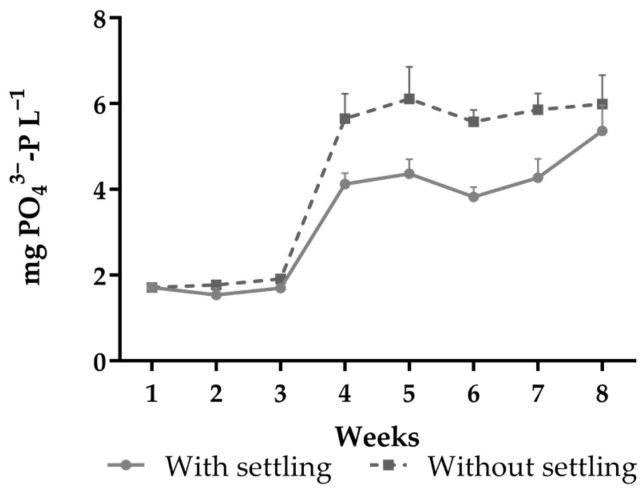
Dissolved orthophosphate in tanks of *Penaeus vannamei* grown in an aquaponic system with and without settling.

**Figure 4 animals-15-01294-f004:**
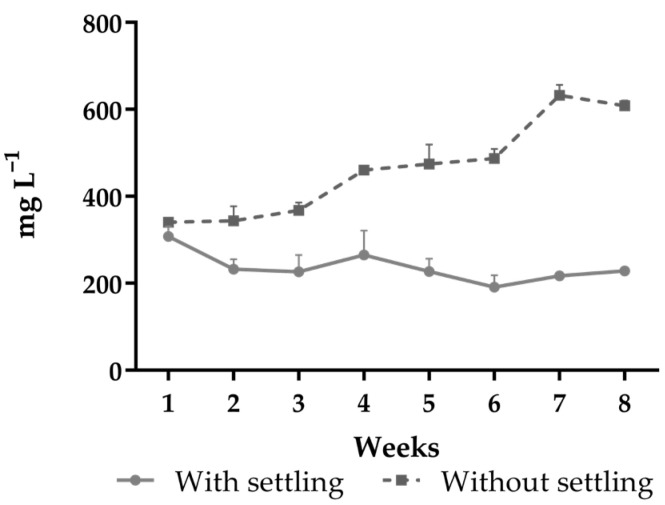
Total suspended solids concentration in the water of *Penaeus vannamei* tanks grown in aquaponic system with and without the use of a settling chamber for eight weeks.

**Table 1 animals-15-01294-t001:** Production indices of *Penaeus vannamei* and *Salicornia neei* cultivated in an aquaponic system with and without the use of a settling chamber for 54 days. Data are presented as a mean ± standard deviation.

Parameters	Treatment		*p*-Value
	With Settling	Without Settling	
Performance of *P. vannamei*			
Survival (%)	88.7 ± 6.7	86.8 ± 5.1	0.6745
Mean initial weight (g)	1.2 ± 0.1	1.2 ± 0.1	0.4683
Mean final weight (g)	10.2 ± 0.3	10.1 ± 0.3	0.6431
Weekly weight gain (g week^−1^)	1.17 ± 0.04	1.15 ± 0.04	0.6283
Final biomass (kg tank^−1^)	2.71 ± 0.14	2.62 ± 0.07	0.3936
Productivity (kg m^−3^)	3.04 ± 0.16	3.27 ± 0.09	0.1125
Total feed input (kg)	3.60 ± 0.05	3.63 ± 0.05	0.4713
Feed conversion ratio (FCR)	1.54 ± 0.11	1.61 ± 0.07	0.3757
Performance of *S*. *neei*			
Mean final weight (g)	14.5 ± 5.9	12.8 ± 4.8	0.3597
Final biomass (g tank^−1^)	383.3 ± 58.4	338.2 ± 33.0	0.4510
Survival (%)	93.8 ± 1.8	94.0 ± 2.1	0.6964
Productivity (kg m^−2^)	1.26 ± 0.24	1.13 ± 0.11	0.4510
*Shrimp plus Salicornia*			
Total final biomass (kg)	3.09 ± 0.11	2.95 ± 0.04	0.1703
Total yield (kg m^−3^)	3.47 ± 0.13	3.65 ± 0.06	0.1096

**Table 2 animals-15-01294-t002:** Water quality variables in the tanks of *Penaeus vannamei* cultivated in aquaponics with and without the use of a settling chamber for 54 days, with a stocking density of 375 shrimp m^−3^.

Parameter	Treatment		RM ANOVA		
	With Settling	Without Settling	T	D	T × D
Salinity (psu)	33.2 ± 1.6	32.5 ± 2.3	ns	<0.0001	0.0030
	(30.0–35.8)	(28.7–35.7)			
Temperature (°C)	28.7 ± 0.9	28.9 ± 0.8	ns	0.0010	ns
	(27.3–32.0)	(27.4–32.0)			
Dissolved oxygen (mg L^−1^)	5.9 ± 0.3	5.8 ± 0.3	ns	0.0020	ns
	(5.0–7.6)	(5.1–7.3)			
pH	8.11 ± 0.07	8.00 ± 0.12	0.0013	0.0006	<0.0001
	(7.99–8.30)	(7.72–8.23)			
Alkalinity (mg CaCO_3_ L^−1^)	164.8 ± 18.8	138.1 ± 16.6	0.0043	0.0091	<0.0001
	(132–212)	(112–168)			
Total ammonia nitrogen (mg L^−1^)	0.49 ± 0.15	0.22 ± 0.16	0.0028	ns	0.0034
	(0.20–0.84)	(0.07–1.09)			
Nitrite (mg NO_2_-N L^−1^)	0.68 ± 0.49	0.13 ± 0.06	0.0047	0.0368	<0.0001
	(0.01–2.25)	(0.01–0.30)			
Nitrate (mg NO_3_-N L^−1^)	6.09 ± 1.49	12.28 ± 6.76	0.0025	0.0016	<0.0001
	(2.92–8.94)	(2.76–28.92)			
Orthophosphate (mg PO_4_^3−^-P L^−1^)	3.48 ± 1.46	4.35 ± 1.95	0.0055	0.0016	0.0101
	(1.33–6.26)	(1.68–6.72)			
TSS (mg L^−1^)	230.0 ± 39.8	464.2 ± 118.2	<0.0001	0.0001	<0.0001
	(140–350)	(287–717)			
VSS (%)	27.7 ± 5.9	40.1 ± 4.7	0.0025	ns	0.0071
	(4.5–47.5)	(29.3–48.0)			
Settleable solids (mL L^−1^)	0.09 ± 0.17	9.17 ± 8.02	<0.0001	<0.0001	<0.0001
	(0.0–1.0)	(0.3–27.0)			

Data are the mean ± standard deviation (minimum–maximum). ns—not significant; T—treatment; D—days; T × D—interaction between treatment and days of culture.

**Table 3 animals-15-01294-t003:** Total sludge produced, total suspended solids (TSS), volatile suspended solids (VSS), fixed suspended solids (FSS), and settleable solids in the eighth week of the trial for the cultivation of *Salicornia neei* and *Penaeus vannamei* in aquaponics with and without the use of settling chamber.

Parameter	Treatment		*p*-Value
	With Settling	Without Settling	
TSS (mg L^−1^)	231.3 ± 10.9	608.5 ± 94	<0.0001
VSS (%)	31.3 ± 2.0	46.1 ± 2.0	0.0012
FSS (%)	72.3 ± 5.9	59.9 ± 4.7	0.0012
Settleable solids (mL L^−1^)	0.08 ± 0.08	19.17 ± 6.15	0.0095
Total sludge produced (g tank^−1^)	122.3 ± 17.3	905.5 ± 101.8	0.0046

Data are presented as a mean ± standard deviation.

## Data Availability

All the data obtained during this study will be made available upon request.

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
