# Peer review of "The Effect of the Use of a Settling Chamber in the Cultivation of Penaeus vannamei and Salicornia neei in Aquaponics with Bioflocs"

_animals, 2025, doi:10.3390/ani15091294_

Round 1
Reviewer 1 Report
Comments and Suggestions for Authors
Such experiment has to run for longer than only 58 days
Author Response
Comments 1: Such experiment has to run for longer than only 58 days.
Response 1: Thank you for your comment. Normally, our experiments are designed to last 8 weeks–this is enough time to reach the commercial size of Penaeus vannamei in Brazil, that varies from 10 g to 15 g (Valenti et al., 2021; Pasco, 2024). Our goal was to reach at least 12-13 g after 54 days, but our high stocking density (375 shrimp.m-3) most probably impaired the growth rate. However, we recognize that we should have carried out the experiment for longer, to better evaluate the effects of the settling chamber and the aquaponic system on cultivation. We will take this into account when designing future experiments.
Valenti, W. C. et al.. Aquaculture in Brazil: past, present and future. Aquaculture Reports. Vol. 19 (2021). https://doi.org/10.1016/j.aqrep.2021.100611
Pasco, J.M. Manejo prático e bem-sucedido na produção de camarão marinho: assessorando fazendas de camarão a utilizar manejos simples e científicos para alcançar bons índices produtivos. Revista da ABCC. Vol. 2 (2024). https://abccam.com.br/wp-content/uploads/2024/08/Abcc_Agosto_Revista_2024_web-2-1.pdf (In Portuguese).
Reviewer 2 Report
Comments and Suggestions for Authors
Overall comments: Solids build-up around the plant roots in the aquaponics system tends to increase as the plants grow (more root structure to capture solids) and as the density of the cultured organisms increase (more food and waste). The current study shows no effect on shrimp and plant performance over the 54 day study with average shrimp size at ~10 grams. The reader might be more interested (concerned) on how the trial will progress as you take the salicornia and shrimp to market size. This is especially true as the TSS in the system without settling reached its "control level" of 600 mg/L only a ~week before the experiment was terminated (i.e. how will the plants perform at the ~600 mg/L level as opposed to gradually increasing to 600?). Since this study has such a strong applied basis why was the trial not conducted until market size of the shrimp was reached?
Specific comments:
Lines 19-21: I would add "through 54-days" after system performance.
Lines 78-80: The authors mention the potential toxicity of nitrate and phosphates build-up in these systems due to the lack of water exchange. In the experiment the nitrate and phosphate were significantly lower in the system with settling. If nutrient build-up is a concern, and there were no differences in performance over 54-days this appears to be a potential benefit unless the authors believe the implementation of the settling chamber outweighs the water quality aspect.
Lines 89-90: While settling chambers can be necessary it's only under certain specific situations which is why settling chambers are typically employed in aquaponic systems.
Lines 146-148: Since breaking down the biofloc to smaller particles is a concern why was the settled biofloc pumped back as opposed to designing a system where the settled biofloc gravity flowed back to the system via gravity?
Lines 245-246: You mention you only removed excess solids once, almost at the end of the trial. Is it not likely excess solids would need to be removed on a ~weekly basis from this point on through commercial harvest size of the shrimp?
Line 261: "evince" should be demonstrated?
Lines 310-303: It would have been nice to see the experiment carried out to harvestable size to fully conclude the benefits or detriment of settling chambers on shrimp and salicornia performance.
Line 328: Would add "through 54-days of production" after plants and shrimp.
Author Response
Comments 1: Overall comments: Solids build-up around the plant roots in the aquaponics system tends to increase as the plants grow (more root structure to capture solids) and as the density of the cultured organisms increase (more food and waste). The current study shows no effect on shrimp and plant performance over the 54 day study with average shrimp size at ~10 grams. The reader might be more interested (concerned) on how the trial will progress as you take the salicornia and shrimp to market size. This is especially true as the TSS in the system without settling reached its "control level" of 600 mg/L only a ~week before the experiment was terminated (i.e. how will the plants perform at the ~600 mg/L level as opposed to gradually increasing to 600?). Since this study has such a strong applied basis why was the trial not conducted until market size of the shrimp was reached?
Response 1: Thank you very much for your comment. Normally, our experiments are designed to last 8 weeks (Poli et al., 2019; Pinheiro et al., 2020)–this is enough time to reach the commercial size of Penaeus vannamei in Brazil, that varies from 10 g to 15 g (Valenti et al., 2021; Pasco, 2024). Our goal was to reach at least 12-13 g after 54 days, but our high stocking density (375 shrimp.m-3) most probably impaired the growth rate. Our concern was also with the survival rate, since we were getting close to the carrying capacity of our experimental units. Although not demonstrate in our data, the Salicornia had reached the commercial size of around 20 cm of shoot length. Based on these factors, we decided to harvest the system after the 54 days initially planned. However, we recognize that we should have carried out the experiment for longer, to better evaluate the effects of the settling chamber and the aquaponic system on cultivation. We will take this into account when designing future experiments.
Comments 2: Lines 19-21: I would add "through 54-days" after system performance.
Response 2: We agree and added the information.
Comments 3: Lines 78-80: The authors mention the potential toxicity of nitrate and phosphates build-up in these systems due to the lack of water exchange. In the experiment the nitrate and phosphate were significantly lower in the system with settling. If nutrient build-up is a concern, and there were no differences in performance over 54-days this appears to be a potential benefit unless the authors believe the implementation of the settling chamber outweighs the water quality aspect.
Response 3: Thank you for pointing this out. We agree and, in order to avoid misleading information regarding our data, decided to remove this sentence from the introduction and reorganize the paragraph. Using the term "toxicity" was not really in line with our data set, since, although there were differences between the treatments, both nitrate and orthophosphate concentrations were well below the safe limit for Penaeus vannamei (Prangnell et al., 2019; Prates et al., 2024).
Comments 4: Lines 89-90: While settling chambers can be necessary it's only under certain specific situations which is why settling chambers are typically employed in aquaponic systems.
Response 4: We agree and improved the text for better understanding.
Comments 5: Lines 146-148: Since breaking down the biofloc to smaller particles is a concern why was the settled biofloc pumped back as opposed to designing a system where the settled biofloc gravity flowed back to the system via gravity?
Response 5: This was our first idea when we designed our aquaponics system. However, we had some construction issues to consider, especially the heights of the shrimp tank and the settling chamber, the slope we would need to transfer the water from the settling tank to the irrigation channels, the need to use a stronger pump to compensate for the height of the water entering the settling chamber, and the use of a solenoid valve to release the solids from time to time. All these factors were analyzed, since they would affect the functioning of the settling chamber too. Thus, that is why we opted for a more "simple" approach for the system, with the use of the pump for the solids.
Comments 6: Lines 245-246: You mention you only removed excess solids once, almost at the end of the trial. Is it not likely excess solids would need to be removed on a ~weekly basis from this point on through commercial harvest size of the shrimp?
Response 6: Yes, we did the first removal of solids in the seventh week of the experiment and, according to our knowledge, it is very likely that these removals would be weekly or at least every two weeks due to the increase in biomass in the tank, especially if we consider the harvest size to be between 25 and 30 g. However, as mentioned in the first comment, the harvest size in Brazil is much smaller, and this was one of the reasons why we decided to end the experiment after 8 weeks. But we understood that issues with the management of solids in the system would probably start at this stage. We will consider this in future experiments.
Comments 7: Line 261: "evince" should be demonstrated?
Response 7: We made the change for a better understanding.
Comments 8: Lines 310-303: It would have been nice to see the experiment carried out to harvestable size to fully conclude the benefits or detriment of settling chambers on shrimp and salicornia performance.
Response 8: Thank you for your comment. We will keep this idea for a future trial.
Comments 9: Line 328: Would add "through 54-days of production" after plants and shrimp.
Response 9: We agree and added the information.
References:
Pasco, J.M. Manejo prático e bem-sucedido na produção de camarão marinho: assessorando fazendas de camarão a utilizar manejos simples e científicos para alcançar bons índices produtivos. Revista da ABCC. Vol. 2 (2024). https://abccam.com.br/wp-content/uploads/2024/08/Abcc_Agosto_Revista_2024_web-2-1.pdf (In Portuguese).
Pinheiro, I. et al. Aquaponic Production of Sarcocornia ambigua and Pacific White Shrimp in Biofloc System at Different Salinities. Aquaculture Vol. 519, (2020). doi:10.1016/j.aquaculture.2019.734918.
Poli et al. Integrated Multitrophic Aquaculture Applied to Shrimp Rearing in a Biofloc System. Aquaculture Vol. 511. (2019). doi.org/10.1016/j.aquaculture.2019.734274
Prangnell, D. I.; Samocha, T. M.; Staresinic, N. Water. In: Sustainable Biofloc Systems for Marine Shrimp. [s.l.] Elsevier, 2019. p. 37–58.
Prates, et al. Determination of acute toxicity and evaluation of the chronic effect of nitrate on compensatory growth of Litopenaeus vannamei reared in a biofloc technology system. Aquaculture. Vol 587. (2024). doi.org/10.1016/j.aquaculture.2024.740862
Valenti, W. C. et al. Aquaculture in Brazil: past, present and future. Aquaculture Reports. Vol. 19 (2021). doi.org/10.1016/j.aqrep.2021.100611
Reviewer 3 Report
Comments and Suggestions for Authors
- Tables 1 and 2 present multiple key datasets, but the manuscript does not clearly specify whether corrections for multiple comparisons (e.g., Bonferroni method) were applied. The authors are advised to provide detailed descriptions of the statistical methods used to control for Type I errors (false positives) and ensure the reliability of the conclusions.
- The experimental results from Week 8 show a significant increase in the standard deviation of NOâ‚‚ concentration in the "with settling" treatment group. The underlying reasons for this anomaly require further clarification. Additionally, the authors should discuss whether the 54-day experimental duration is potentially linked to this fluctuation in water quality parameters.
- Although the study observed elevated TAN and NOâ‚‚ levels in the settling treatment, the physiological effects of these changes on cultured organisms (e.g., shrimp) were not thoroughly explored. The authors are encouraged to expand the discussion on how these critical water quality parameters, particularly the toxicity threshold of NOâ‚‚, may influence long-term shrimp health, growth performance, or survival rates.
- While the study concludes that "continuous use of a settling chamber is unnecessary," it remains unclear whether this finding applies to higher stocking densities or different salinity conditions. The authors should elaborate on this point in the Discussion section, clarifying the scope of applicability for their conclusions or identifying conditions that warrant further validation.
- The manuscript does not systematically address the limitations of this study, such as the influence of environmental controls or experimental duration on the results. The authors are recommended to include such discussions and explicitly propose follow-up studies under more rigorously controlled conditions to verify the robustness of the current findings.
Author Response
Comments 1: Tables 1 and 2 present multiple key datasets, but the manuscript does not clearly specify whether corrections for multiple comparisons (e.g., Bonferroni method) were applied. The authors are advised to provide detailed descriptions of the statistical methods used to control for Type I errors (false positives) and ensure the reliability of the conclusions.
Response 1: Thank you for pointing this out. Yes, we applied the Bonferroni test, but unfortunately forgot to add this information to the material and methods. We made the correction.
Comments 2: The experimental results from Week 8 show a significant increase in the standard deviation of NOâ‚‚ concentration in the "with settling" treatment group. The underlying reasons for this anomaly require further clarification. Additionally, the authors should discuss whether the 54-day experimental duration is potentially linked to this fluctuation in water quality parameters.
Response 2: We believe that the increase in the nitrite concentration is linked to the low microbial abundance in the tanks of the treatment with settling, leading to the instability of the nitrification process and, consequently, accumulation of nitrogen compounds in the water. We reformulated the paragraph in the discussion for better understanding of this effect.
Comments 3: Although the study observed elevated TAN and NOâ‚‚ levels in the settling treatment, the physiological effects of these changes on cultured organisms (e.g., shrimp) were not thoroughly explored. The authors are encouraged to expand the discussion on how these critical water quality parameters, particularly the toxicity threshold of NOâ‚‚, may influence long-term shrimp health, growth performance, or survival rates.
Response 3: Although the concentrations of TAN and NO2 were higher in the treatment with settling, it remained below the limit tolerated by P. vannamei throughout the experimental period of 8 weeks. It is not clear to us, however, whether the nitrite would continue to accumulate as the experiment went on, for example until the shrimp reached a weight of 25 g as suggested by the other reviewers. This is something we will have to take into account in future experiments.
Comments 4: While the study concludes that "continuous use of a settling chamber is unnecessary," it remains unclear whether this finding applies to higher stocking densities or different salinity conditions. The authors should elaborate on this point in the Discussion section, clarifying the scope of applicability for their conclusions or identifying conditions that warrant further validation.
Response 4: We did the first removal of solids in the seventh week of the experiment and, according to our knowledge, it is very likely that these removals would be weekly or at least every two weeks due to the increase in biomass in the tank, especially if we consider the harvest size to be between 25 and 30 g. However, the harvest size in Brazil is much smaller, and this was one of the reasons why we decided to end the experiment after 8 weeks. We recognize that we should have carried out the experiment for longer, to better evaluate the effects of the settling chamber and the aquaponic system on cultivation. We will take this into account when designing future experiments. Regarding the salinity, this is something we will have to evaluate in a separate experiment.
Comments 5: The manuscript does not systematically address the limitations of this study, such as the influence of environmental controls or experimental duration on the results. The authors are recommended to include such discussions and explicitly propose follow-up studies under more rigorously controlled conditions to verify the robustness of the current findings.
Response 5: Thank you for pointing this out. We added more information regarding this on the discussion.
Reviewer 4 Report
Comments and Suggestions for Authors
see attached for modifications
I insist on the conclusion being that during the first 2 months you do not need a settling. but after that period removal of solids will be needed. Maybe settling chamber can be ommited for pregrow of the shrimp, but not after until commercial size.

Author Response
- Summary
Thank you very much for taking the time to review this manuscript. Please find the detailed responses below and the corresponding revisions/corrections highlighted/in track changes in the re-submitted files.
- Point-by-point response to Comments and Suggestions for Authors
Comments 1: Line 20: you must say here that your results are valid only in first 2 months of cultivation. "use of a settling chamber is not essential for system performance in the 2 first months"
Response 1: We agree and changed the sentence.
Comments 2: Line 38-39: same !! You might want to say that during pre-grow settling chamber can be omitted. Otherwise you mislead farmers.
Response 2: We agree and added the information.
Comments 3: Would you like to add aquaponics.
Response 3: We decided not to use it as a keyword since it is already on the title.
Comments 4: Line 61-62: in reality, BFT does NOT replace hydroponics. You might want to change this sentence and remove the word hydroponics. Sentence would be like that: ..., combining the components of solids removal and nitrification in a single unit ...
Response 4: We agree and improved the text.
Comments 5: Line 63: the word Hence in unnecessary.
Response 5: We agree and removed the word.
Comments 6: Line 65-71, I would like to see here more clearly explains that BFT "replaces" solids removal and nitrification.
Response 6: We improved the text for better understanding.
Comments 7: Line 78-82: are in the subject of nutrients. I would move them upwards, just after ref [15-19] in line 74. Doing that, you keep a continuous flow of information on the subject of solids, with current lines 77- 78, then 74-76, then 83-
Response 7: Thank you for pointing this out. We altered the order of the paragraph and added new information.
Comments 8: Line 89-90 is bizarre. Do you mean that in aquaponics with BFT, the use of settling chamber ...
Response 8: We agree and improved the text.
Comments 9: Line 97, I recommend you insist more with the fact that plant roots are affected by excessive solids [7].
Response 9: We agree and improved the text.
Comments 10: Line 99: the word continuous here is a bit misleading. Because earlier line 86 you say that continuous may be detrimental. I would remove it.
Response 10: We agree and removed the word.
Comments 11: Line 245: I do not understand what you mean. It seems that without settling is predominantly chemautotrophic and therefore solids are not an issue until the very end. I don't see the logic here. Please reformulate.
Comments 12: I would reorder with : the use of the settling chamber in this treatment was necessary almost at the end of the trial. And Line 246 I suggest you remove the detail (almost at the end of the trial)
Responses 11 and 12: We improved the text for better understanding.
Comments 13: Line 256: to improve, I suggest : The reduction in the concentration of total suspended solids and settable solids in this condition may also be responsible for decreasing ... of course you remove “This condition led” in line 256 and you remove “in the water” line 257
Response 13: We agree with the suggestion and made the changes in the sentence.
Comments 14: Line 261: the verb evince is not clear for me, and probably for others! Please replace
Response 14: We agree and replaced the word.
Comments 15: Somewhere in the discussion you may want to say that in a way, settling solids is detrimental to BFT. Or you loose the benefit of BFT by using settling chamber. See where you can add that with better words than me.
Response 15: Thank you very much for this suggestion. We added new information to the text.
Comments 16: Line 300-301 there is no verb in the sentence !
Response 16: We improved the text for better understanding.
Comments 17: Line 322: you mention that plants did reach the commercial size. Please add a sentence on shrimp size. To my knowledge, vannamei is sold at 20-25g. If plants reach commercial size before solids are too high, how is it for shrimps: will you change water to eliminate solids and plant another generation of plants in the same system, or 7 if the farmer has no settling chamber, how can he continue the growing process of shrimp?
Response 17: Thank you very much for your comment. Normally, our experiments are designed to last 8 weeks (Poli et al., 2019; Pinheiro et al., 2020)–this is enough time to reach the commercial size of Penaeus vannamei in Brazil, that varies from 10 g to 15 g (Valenti et al., 2021; Pasco, 2024). Although not demonstrate in our data, the Salicornia had reached the commercial size of around 20 cm of shoot length. It is very likely that solids removals would be weekly or at least every two weeks due to the increase in biomass in the tank, especially if we consider the harvest size to be between 25 and 30 g. In case Salicornia reaches the commercial size before the shrimp, the producer can have another production cycle of salicornia in the system, through partial harvest (keeping the root system in the system for regrowth). The solutions in case the producer has no settling tank are to perform water exchange, to filter the solids with a filtering bag or even split the shrimp biomass into another tank, using the stable biofloc as an inoculum to start the second tank.
Comments 18: Line 324-325: P. vannamei in bioflocs and S. neei in aquaponics
Response 18: We consider the aquaponics with bioflocs as a whole system, to produce both animals and plants, but we made changes to the text for better understanding.
Comments 19: Line 330: the end of the sentence should cut and be repositioned in line 328. => ... did not hinder the growth of plants and shrimps, and the aquaponics structure is simplified. However, the use for the cultivation of marine shrimp.
Response 19: We agree with the suggestion and changed the sentence.
Comments 20: Line 330: the end of the sentence should cut and be repositioned in line 328. => ... did not hinder the growth of plants and shrimps, and the aquaponics structure is simplified. However, the use for the cultivation of marine shrimp.
I am a bit puzzled here. Because in reality the farmer will need to invest in a settling chamber even for using it every month or so!! For me, BFT and aquaponics do simplify because you don't need a biofilter. But you need a settling chamber.
I believe also somewhere in conclusion you could repeat that you did not change water at all. That is of interest for the farmers in reducing costs.
Response 20: We agree with the suggestions and made significant changes to the conclusion for better understanding.
Reviewer 5 Report
Comments and Suggestions for Authors
The present study was designed to evaluate the effect of continuous solids removal using a settling chamber on the cultivation of Penaeus vannamei and Salicornia neei in a biofloc- based aquaponic system. Two treatments were compared: one with a settling chamber and one without. In the settling treatment, water was first directed through the chamber before reaching the hydroponic bench, while in the treatment without settling, water was pumped directly to the plants. Over a 54-day trial, the settling treatment resulted in reduced suspended solids but increased TAN and NO₂ levels, while NO₃ concentrations remained stable. Despite differences in water quality parameters, no significant variations were observed in shrimp or plant production between treatments. The research work is systematic and meticulous. However, there are still some problems that need to be corrected in the article.
Major comment
- Introduction It is important to emphasise that this study is the first to combine Salicornia neei with Penaeus vannamei in a biofloc hydroponic system and to explore the effects of continuous use of sedimentation tanks. Much of the existing literature focuses on freshwater systems or non-saline plants, and there is a need to clarify the unique contribution of saltwater environments.
- add controversial studies on the role of sedimentation tanks in biofloc hydroponic systems (e.g., whether they destabilise microbial communities), citing key literature to highlight the need for research.
- the methods need to state whether three replications satisfy statistical requirements (e.g., efficacy analysis) and describe the randomisation process (e.g., whether light and temperature gradients were controlled in the greenhouse) to enhance the credibility of the experimental design.
- add the effect of hydraulic retention time (HRT) in the settling tank, the flow rate of pumped back solids (15 L min-¹) on floc structure, and why the 30-minute pump back frequency was chosen (citation of similar studies is required to support this).
- clarify whether the sampling time for water quality parameters (e.g., TAN, NOâ‚‚) is fixed (e.g., same time each day) and discuss the effect of diurnal variations on the results (e.g., fluctuations in dissolved oxygen).
- Present the time series data for the water quality parameters (TAN, NOâ‚‚, NO₃) in line graphs rather than tables to visualise the stability advantage of the ‘no settling tank treatment’ (e.g. Figure 2 can be broken down into graphs).
- Specific results of the Shapiro-Wilk and Levene tests (e.g., p-value ranges) were described in the methodology, and it was confirmed whether the repeated measures ANOVA corrected for the assumption of sphericity (e.g., Greenhouse-Geisser adjustment).
- explain the water quality-yield paradox
Discuss why elevated TAN/NOâ‚‚ did not affect shrimp and plant yields, which needs to be combined with mechanisms of ammonia tolerance in salthorn grasses (e.g., ionic regulating ability of saline plants in the literature) and nutrient compensation by bioflocs.
8 Controversy over the role of sedimentation tanks
In relation to previous studies (e.g. sedimentation tanks may reduce nitrification efficiency), potential reasons why sedimentation tanks did not significantly alter yields in this study were analysed (e.g. plant uptake to counteract negative effects).
- Conclusions need to emphasise that the omission of sedimentation tanks reduces system complexity and cost, and quantify the potential benefits (e.g. XX% reduction in energy consumption), providing a replicable solution for saline agriculture.
- It is recommended to add the effect of long term trials (>54 days) on sludge accumulation or system performance at different salinities/plant densities to enhance the depth of the study.
- ‘Mean final weight (g)’ in Table 1 should be retained to one decimal place (e.g. 10.2→10.2±0.3), in line with other data.
- Standardise terminology: use ‘settling chamber’ rather than ‘sedimentation chamber’ throughout (now mixed).
- Check the unit format (e.g. ‘μmol photons m-² s-¹’ should be in block letters).
- Additional keywords: add ‘nutrient recycling’ or ‘saline aquaponics’ to improve the relevance of the search.
- Optimisation of diagrams: Figure 1 needs to be scaled and the difference in flowcharts ‘with/without settling’ (e.g. direction of arrows) needs to be differentiated.
The English could be improved to more clearly express the research.
Author Response
Comments 1: Introduction It is important to emphasise that this study is the first to combine Salicornia neei with Penaeus vannamei in a biofloc hydroponic system and to explore the effects of continuous use of sedimentation tanks. Much of the existing literature focuses on freshwater systems or non-saline plants, and there is a need to clarify the unique contribution of saltwater environments.
Response 1: Thank you for pointing this out. We added more information on the introduction.
Comments 2: add controversial studies on the role of sedimentation tanks in biofloc hydroponic systems (e.g., whether they destabilise microbial communities), citing key literature to highlight the need for research.
Response 2: We added more information on the introduction.
Comments 3: the methods need to state whether three replications satisfy statistical requirements (e.g., efficacy analysis) and describe the randomisation process (e.g., whether light and temperature gradients were controlled in the greenhouse) to enhance the credibility of the experimental design.
Response 3: The experimental units were randomly distributed in the greenhouse. Light and temperature inside of the greenhouse were not controlled, they followed the natural pattern of the period of the experiment (spring in southern Brazil). Light intensity was measured daily at noon, above each of the hydroponic benches.
Comments 4: add the effect of hydraulic retention time (HRT) in the settling tank, the flow rate of pumped back solids (15 L min-¹) on floc structure, and why the 30-minute pump back frequency was chosen (citation of similar studies is required to support this).
Response 4: When we first designed our experimental units, back in 2014, we did some pre-trials on the functioning of the settling chamber and the ideal inlet and outlet flows for the structure that we had available. Therefore, we had to consider an outlet flow to pump enough volume where all the solids settle on the conical bottom of the chamber would return to the tank, but also had to take into account the inlet flow, that means, how long it would take for the settling tank to be full again. Based on these results, we set the minimum flow of 15 L/min for the sludge return and 3 L/min for the inlet. On our very first work (you can check Pinheiro et al., 2017) the sludge was pumped every hour, but we noticed that maybe there was denitrification occurring inside of the settling chamber due to low oxygen levels at the bottom. That’s why we decided to decrease this time to every 30 min – however the same pattern was still observed.
Comments 5: clarify whether the sampling time for water quality parameters (e.g., TAN, NOâ‚‚) is fixed (e.g., same time each day) and discuss the effect of diurnal variations on the results (e.g., fluctuations in dissolved oxygen).
Response 5: Yes, all the samples/measurements were taken at the same time each day. Dissolved oxygen and temperature were monitored twice daily (8 am and 5 pm) using a YSI oximeter. Every Monday and Thursday at 8 am (before feeding), we collected the water samples for the analyzes in the lab (salinity, pH, alkalinity, TAN, and nitrite). On Thursdays, we also analyzed nitrate and orthophosphate.
We didn’t observe any significant diurnal fluctuations in oxygen and temperature (table below) that can be discussed.
|
|
With settling |
Without settling |
|
DO morning (mg.L-1) |
5.9 ± 0.2 |
5.8 ± 0.3 |
|
DO afternoon (mg.L-1) |
5.9 ± 0.3 |
5.8 ± 0.3 |
|
Temperature morning (°C) |
28.1 ± 0.4 |
28.5 ± 0.4 |
|
Temperature afternoon (°C) |
29.1 ± 1.0 |
29.2 ± 0.9 |
Comments 6: Present the time series data for the water quality parameters (TAN, NOâ‚‚, NO₃) in line graphs rather than tables to visualise the stability advantage of the ‘no settling tank treatment’ (e.g. Figure 2 can be broken down into graphs).
Response 6: We agree that visualizing this data in graphs is important. We don’t understand this comment, since we have already presented the TAN, NO2, NO3, PO4 and TSS data in line graphs throughout the weeks of cultivation (figures 2, 3, and 4 respectively) on the first version of the manuscript.
Comments 7: Specific results of the Shapiro-Wilk and Levene tests (e.g., p-value ranges) were described in the methodology, and it was confirmed whether the repeated measures ANOVA corrected for the assumption of sphericity (e.g., Greenhouse-Geisser adjustment).
Response 7: We added the information.
Comments 8: explain the water quality-yield paradox
Discuss why elevated TAN/NOâ‚‚ did not affect shrimp and plant yields, which needs to be combined with mechanisms of ammonia tolerance in salthorn grasses (e.g., ionic regulating ability of saline plants in the literature) and nutrient compensation by bioflocs.
Response 8: Thank you for pointing this out. Although the concentrations of TAN were higher in the treatment with settling, it was still under the safe levels for both species. In the case of the plants, Salicornia species can remove ammonium efficiently at concentrations below 4 mM (around 56 mg N L-1), with a maximum uptake rate of 7 mM N g-1 fresh weight. Salinites above 200 mM have been shown to reduce ammonium accumulation in roots and shoots, potentially alleviating toxicity symptoms. This suggests that salinity plays a protective role in ammonium detoxification. We added the info on the discussion.
Comments 9: Controversy over the role of sedimentation tanks
In relation to previous studies (e.g. sedimentation tanks may reduce nitrification efficiency), potential reasons why sedimentation tanks did not significantly alter yields in this study were analysed (e.g. plant uptake to counteract negative effects).
Response 9: We added the info on the discussion.
Comments 10: Conclusions need to emphasise that the omission of sedimentation tanks reduces system complexity and cost, and quantify the potential benefits (e.g. XX% reduction in energy consumption), providing a replicable solution for saline agriculture.
Response 10: Thank you for the comment. We decided not to address cost related advantages since we didn’t perform any economic analysis of the system. However, we improved the text of the conclusion to highlight our findings in a better way.
Comments 11: It is recommended to add the effect of long term trials (>54 days) on sludge accumulation or system performance at different salinities/plant densities to enhance the depth of the study.
Response 11: Thank you for mentioning this. We added the info in the discussion.
Comments 12: ‘Mean final weight (g)’ in Table 1 should be retained to one decimal place (e.g. 10.2→10.2±0.3), in line with other data.
Responses 12: We made the correction.
Comments 13: Standardise terminology: use ‘settling chamber’ rather than ‘sedimentation chamber’ throughout (now mixed).
Response 13: Thank you for pointing this out. We checked the whole document and it is all standardized.
Comments 14: Check the unit format (e.g. ‘μmol photons m-² s-¹’ should be in block letters).
Response 14: We made the correction.
Comments 15: Additional keywords: add ‘nutrient recycling’ or ‘saline aquaponics’ to improve the relevance of the search.
Response 15: We agree and added the keywords.
Comments 16: Optimisation of diagrams: Figure 1 needs to be scaled and the difference in flowcharts ‘with/without settling’ (e.g. direction of arrows) needs to be differentiated.
Response 16: Figure 1 only serves to illustrate the system. We’ve drawn it as close to the real scale as possible but, unfortunately, it’s not possible to add it to the diagram as it wouldn’t be completely accurate.
Round 2
Reviewer 5 Report
Comments and Suggestions for Authors
no comments
Author Response
We would like to thank the reviewer for his/her valuable insights on our work.